# Research on the Relationship between the Amylopectin Structure and the Physicochemical Properties of Starch Extracted from Glutinous Rice

**DOI:** 10.3390/foods12030460

**Published:** 2023-01-18

**Authors:** Bingqing Wang, Jialu Xu, Dandan Guo, Changzhi Long, Zhongxin Zhang, Ying Cheng, Huiying Huang, Peng Wen, Haohua He, Xiaopeng He

**Affiliations:** Key Laboratory of Crop Physiology, Ecology and Genetic Breeding, Ministry of Education, College of Agronomy, Jiangxi Agricultural University, Nanchang 330045, China

**Keywords:** glutinous rice, amylopectin structure, starch physicochemical properties, gelatinization temperature

## Abstract

Glutinous rice has very low amylose content and is a good material for determining the structure and physicochemical properties of amylopectin. We selected 29 glutinous rice varieties and determined the amylopectin structure by high-performance anion exchange chromatography with the pulsed amperometric detection method. We also determined the correlation between amylopectin structure and the physicochemical properties of starch extracted from these varieties. The results showed that the amylopectin chain ratio Σdegree of polymerization (DP) ≤ 11/ΣDP ≤ 24 of 29 glutinous rice varieties was greater than 0.26, signifying that these varieties contained type II amylopectin. The results of the correlation analysis with gelatinization temperature showed that ΣDP 6–11 was significantly negatively correlated with the onset gelatinization temperature (GT) (T_O_), peak GT (T_P_), and conclusion GT (T_C_). Among the thermodynamic properties, ΣDP 12–24 was significantly positively correlated with To, Tp, and Tc, ΣDP 25–36 was significantly negatively correlated with To, Tp, and Tc, and ΣDP ≥ 37 had no correlation with the thermodynamic properties. The results of correlation analysis with RVA spectrum characteristic values showed that ΣDP 6–11 was significantly negatively correlated with hot paste viscosity (HPV), cool paste viscosity (CPV), consistency viscosity (CSV), peak time (PeT), and pasting temperature (PaT) among the Rapid Visco Analyzer (RVA) profile characteristics, ΣDP 12–24 was significantly positively correlated with HPV, CPV, CSV, PeT, and PaT, and ΣDP ≥ 25 had no correlation with the viscosity characteristics. Therefore, we concluded that the amylopectin structure had a greater effect on the T_O_, T_P_, T_C_, ΔH and peak viscosity, HPV, CPV, CSV, PeT, and PaT. The glutinous rice varieties with a higher distribution of short chains and a lower distribution of medium and long chains in the amylopectin structure resulted in lower GT and RVA spectrum characteristic values.

## 1. Introduction

The structure of amylopectin has an important effect on the physicochemical properties of starch [1,2], which is the main reason for the quality difference between rice varieties with similar amylose contents (AC) [3]. Previous studies have shown that the amylopectin structure has a significant effect on rice gelatinization temperature (GT) [4]. Nishi et al. [5] found that the decrease in the short chain of amylopectin leads to an increase in GT. Vandeputte et al. [2] and He [6] found that the short chain and the medium-long chain in amylopectin were significantly negatively and significantly positively correlated with rice gelatinization heat, respectively. The relationship between the amylopectin structure and Rapid Visco Analyzer (RVA) spectrum characteristic values has been different in different studies. By performing gel-filtration chromatography, Cai et al. [7] found that FrIII (DP 10–17 glucose unite) in the amylopectin structure was significantly positively correlated with peak viscosity (PKV) and breakdown viscosity (BDV) in the RVA spectrum characteristic values, whereas FrI (DP > 100 glucose unite) and FrII (DP 44–47 glucose unite) showed an opposite trend. Han et al. [8] used the same method to conclude that FrI was significantly negatively correlated with BDV, whereas FrIII was significantly positively correlated with BDV. He [9] used the improved fluorophore-assisted carbohydrate electrophoresis (FACE) method based on a DNA sequencer analysis and found that the proportion of branch chains in different chain length ranges was mainly related to the pasting temperature and the relative crystallinity of starch but was not closely related to the gel consistency and RVA spectrum characteristic values of starch. According to previous studies, we found that AC is not the only determinant of rice quality and the physicochemical properties of starch. Significant differences exist in the physicochemical properties of starch among AC-similar varieties, and the amylopectin structure also has an important effect on rice quality and the physicochemical properties of starch [9]. As the amylopectin content of glutinous rice is very high, accounting for 95–100% of the total starch content, the amylopectin structure of glutinous rice is more complex, which has a great effect on its quality. At the same time, the AC of most glutinous rice is very low (<2%). By using glutinous rice as an experimental material, the effect of amylose can be considerably reduced, and the relationship between the fine structure of amylopectin and the physicochemical properties of starch can be studied more effectively. Although glutinous rice is not suitable for staple food, it is widely used to make traditional food, drinks and condiments, such as tangyuan, rice wine, vinegar, etc. It can be used as raw material of biological composite materials. It is also used in the medical industry (capsule), skin care products, printing, and other fields [10,11]. Therefore, the research on quality breeding of glutinous rice has more potential. In this study, the distribution of amylopectin chain length and the physicochemical properties of amylopectin obtained from 29 varieties of glutinous rice were determined, and the correlation analysis was performed to determine the effect of the crystalline structure of amylopectin from glutinous rice on the physicochemical properties of starch. We aimed to reveal the genetic basis of the formation of amylopectin structure in glutinous rice and provide a theoretical basis for the breeding and improvement of glutinous rice starch quality.

## 2. Materials and Methods

### 2.1. Materials and Instruments

In this study, 29 glutinous rice varieties of different qualities were used as study materials, including 17 varieties of indica rice and 12 varieties of japonica rice (Table 1). Test varieties were planted in the Hainan South Breeding Base of Jiangxi Agricultural University on 5 December 2019. The field was managed routinely, wherein sowing was performed in stages, and the grains were harvested at the same time. After the rice matured, every single plant was harvested and stored for three months. After the quality of the harvest was assessed, the brown rice was obtained after processing the harvest using a three-vertical husker (SY88-TH, Korea Shuanglong Machinery Manufacturing Co., Ltd., Taegu, Republic of Korea), which was then polished using a rice polisher machine (SY2001 + NSART100, Korea Shuanglong Machinery Manufacturing Co., Ltd., Taegu, Republic of Korea) to obtain the polished rice. The polished glutinous rice was ground into rice flour using a cyclone mill (CT293, FOSS, Hilleroed, Denmark) and passed through a 100-mesh sieve. The rice flour was dried in an oven set at 37 °C for 48 h, placed at room temperature (25 °C) for 24 h to maintain its water content at 13 ± 1%, and then sealed for subsequent experiments. All indexes were repeated three times for each sample during determination.

The following experimental instruments were used in this study: TU-1810D ultraviolet-visible spectrophotometer: Beijing Puxi Company, Beijing, China; Thermo ICS5000+ ion chromatography system: Thermo Fisher Scientific, Waltham, MA, USA; Matersizer 3000 Laser Particle Size Analyzer: Malvern Instruments Ltd., Worcestershire, UK; X’Pert Pro X-ray diffractometer: PANalytical, Almelo, The Netherlands; RVA-TecMaster viscometer: Perten, Stockholm, Sweden; and DSC 4000 differential scanning calorimeter (DSC): Perkin-Elmer, Waltham, MA, USA.

### 2.2. Wx Genotyping

DNA was extracted according to the cetrimonium bromide method [12], and sequence-tagged site (STS) markers were developed by Sun et al. [13] for the different sequence in the second exon of the rice waxy (*Wx*) gene to detect glutinous rice varieties. The polymerase chain reaction (PCR) was performed under the following conditions: 5 min at 94 °C, followed by 30 s at 94 °C, 30 s at 55 °C, and 30 s at 72 °C for 35 cycles, and 10 min at 72 °C for a final extension. Only waxy varieties have 492 bp PCR products, while non waxy varieties have no PCR products [13]. PCR amplification primers used are as follows:Glu-F: 5′-GGGTGCAACGGCCAGGAT-3′
        Glu-R: 5′-TGGAACCCGTGGGCTTGA-3′

### 2.3. Determination of AC

AC was determined according to the Chinese national standard GB/T15683-2008 [14].

### 2.4. Extraction and Purification of Starch

Starch was extracted and purified according to the method of Wei et al. [15].

### 2.5. Determination of the Chain-Length Distribution of Starch

According to the method of Zhang et al. [16], the amylopectin chain-length distribution was determined by high-performance anion exchange chromatography with pulsed amperometric detection (HAPED-PAD) using the Thermo ICS5000+ ion chromatography system equipped with pulsed Abe detection (Thermo Fisher Scientific, Waltham, MA, USA). Dionex™ CarboPac™ PA10 (250 × 4.0 mm, 10 µm) liquid chromatographic column is adopted for chromatographic system, with the injection volume of 20 µL. Moving phase A: 200 mM NaOH; Phase B: 200 mM NaOH/200 mM NaAC, the column temperature is 30 °C, and the monosaccharide components are analyzed and detected by electrochemical detector.

### 2.6. Determination of the Relative Crystallinity

X’Pert Pro X-ray diffractometer (PANalytical, Almelo, The Netherlands) was used to analyze the X-ray diffraction patterns of the crystallographic structure of starch. The relative crystallinity of starch was calculated using the MDI Jade software.

### 2.7. Determination of the Physicochemical Properties of Starch

#### 2.7.1. Determination of Starch Viscosity

The starch viscosity was determined using the RVA-TecMaster viscometer (Perten, Stockholm, Sweden) and its supporting software, Thermal Cycle for Windows, according to the American Association of Cereal Chemists operating procedures (1995 61-02) [17]. The obtained results were based on various RVA spectrum characteristic values, including PKV, hot paste viscosity (HPV), cool paste viscosity (CPV), setback viscosity (SBV), BDV, consistency viscosity (CSV), peak time (PeT), and pasting temperature (PaT).

#### 2.7.2. Determination of the Thermodynamic Properties

The thermodynamic properties were measured using a DSC (DSC 4000, Perkin-Elmer, Waltham, MA, USA), and the sample curves were analyzed using the supporting software Pyris Manager. The onset GT (T_O_), peak GT (T_P_), conclusion GT (T_C_), and gelatinization enthalpy (ΔH) were recorded. The specific methods are: weigh 5.0 mg of rice flour sample with 14.0% water content into an aluminum crucible and add 10 μL deionized water, after mixing, use a matching sample press to seal the crucible, and leave it in a 4 °C refrigerator overnight; before the test, take out the crucible stably, place it at room temperature and balance it for 1 h, and then go on the machine for measurement; take the empty disk as the control, and conduct 10 °C/min heating at 30~110 °C; analyze the sample curve and record T_O_, T_P_, T_C,_ and ΔH.

### 2.8. Statistical Analysis

The structural and physicochemical properties of each amylopectin were measured in duplicate. Excel 2011 and IBM Statistical Package for Social Sciences Statistics 22.0 data processing systems were used for analysis of the phenotypic data of the tested varieties and analysis of variance (ANOVA), Duncan’s multiple comparisons, and Pearson’s correlation. The means of duplicated measurements were used for the analysis. Significant differences in the mean values were determined at *p* < 0.05.

## 3. Results

### 3.1. Wx Gene Type and AC of the Glutinous Rice Varieties

Table 1 shows the AC of the tested varieties. The results showed a little difference in AC among the test varieties, with the highest AC in Yunhenuo (2.91%) and the lowest in Anyinuo (0.46%), with an average AC of 2.06%.

The glutinous gene *wx* is an allelic variation of the *Wx* gene, which has a recessive mutation due to 23-bp fragment deletion in the second exon of the *Wx* gene. The appearance of glutinous rice greatly differs from that of non-glutinous rice, but distinguishing this appearance among the heterozygous genotypes is difficult using conventional methods [15]. Therefore, we used the STS dominant molecular markers designed by Sun et al. [13] to detect glutinous rice varieties among the test varieties. The glutinous appearance of the test varieties in this study was due to 23-bp fragment deletion in the second exon of the *Wx* gene, and the amplified fragment length was 492 bp. According to AC determination and the *Wx* genotyping results of the tested varieties, AC had little effect on the physicochemical properties of starch in glutinous rice, and the amylopectin structure was the determining factor affecting the physicochemical properties of starch.

### 3.2. Amylopectin Structure and the Physicochemical Properties of Starch in the Different Glutinous Rice Varieties

#### 3.2.1. Amylopectin Structure

In this study, the amylopectin chain length and its chain-length distribution in the 29 test varieties were determined according to the method of He Xiaopeng [6]. The amylopectin chain lengths were divided into short (DP 6–11), medium-long (DP 12–24), long (DP 25–36), and extra-long (DP ≥ 37) chain lengths (Table 2). Their distribution in the test varieties was different based on their DP [6]. The distribution range of short chains was 18.087–25.332%, among which, Guazixiangdao had the highest and Heixiangnuo had the lowest DP. The short-chain distribution, DP 6–11, in the rice starch branch was the main factor affecting its GT. The short-chain distribution with DP ≤ 11 was higher in most of the glutinous rice varieties than that in the common rice. Therefore, the GTs of the glutinous rice varieties were generally low. The distribution range of ΣDP 12–24 was 52.609–60.592%, among which Zaonuo116 had the highest and Xiangyanuo had the lowest DP. The distribution range of ΣDP 25–36 was 10.86–12.25% and that of DP ≥ 37 was 9.39–11.30%. The average DP was between 19.01 and 20.03, of which Heixiangnuo had the highest and Guazixiangdao had the lowest DP.

#### 3.2.2. Physicochemical Properties of Starch

The results of the physicochemical properties of starch in the test varieties are shown in Table 3 and Table 4. The results showed that the relative crystallinity of starch granules in the test varieties ranged from 31.75% to 40.47%, which belonged to the typical A-type crystalline structure [18]. The relative crystallinity of most glutinous rice varieties was higher than that of the common rice [19].

Among the RVA spectrum characteristic values of the test varieties, PKV ranged from 1523 centipoise (cP) to 3425.33 cP, which was generally lower than that of the common rice. HPV ranged from 882 cP to 2111.33 cP, CPV ranged from 1085 cP to 2563 cP, BDV ranged from 641 cP to 1529 cP, SBV ranged from −1222.67 cP to −363.67 cP, CSV ranged from 203 cP to 484 cP, PeT ranged from 3.76 min to 4.78 min, and PaT ranged from 70.95 °C to 83.00 °C. The measurement results of the thermodynamic characteristics showed that the ranges of variation of T_O_, T_P_, and T_C_ values were 62.54–76.91 °C, 70.25–80.66 °C, and 77.92–88.18 °C, respectively. In the heating process of glutinous rice starch, due to the complexity of amylopectin structure, the pasting temperature of individual varieties is higher. Compared with ordinary rice, most glutinous rice varieties had lower GTs. The ΔH was between 6.33–12.10 J/g, with a mean value of 9.30 J/g. The large difference in ΔH among the test varieties indicated that during the heating process of the glutinous rice starch, due to the complexity of the amylopectin structure, the enthalpy change values among the varieties were different.

### 3.3. Correlation Analysis between the Thermodynamic and Physicochemical Properties of Starch

Further correlation analysis between the thermodynamic and physicochemical properties of starch in different glutinous rice varieties (Table 5) revealed a close correlation between the thermodynamic properties and most of the RVA spectrum characteristic values.

Except that BDV and SBV had no correlation with thermodynamic parameters, and PKV had no correlation with T_O_, T_C_, and ΔH, the other characteristic values were significantly correlated with the thermodynamic parameters. Relative crystallinity and ΔH were significantly positively correlated. To and Tc were significantly positively correlated with HPV, CPV, CSV, PeT, and PaT. Tp was significantly positively correlated with PKV, HPV, CPV, CSV, PeT, and PaT. ΔH was significantly positively correlated with PaT, HPV, CPV, CSV, and PeT. Among these, the correlation coefficients between PaT and T_O_, T_P_, T_C_, and ΔH were the highest, which were 0.965, 0.944, 0.689, and 0.591, respectively, indicating a great correlation among each of the physicochemical properties of starch. PaT in the RVA spectrum can differ in GTs of different varieties to some extent.

### 3.4. Correlation Analysis between the Amylopectin Structure and Physicochemical Properties of Starch

The correlation analysis results of the amylopectin chain length distribution and physicochemical properties of starch are shown in Table 6. ΣDP 6–11 in the test varieties was significantly negatively correlated with PKV, HPV, CPV, CSV, PeT, and PaT in the RVA spectrum characteristic values and T_O_, T_P_, T_C_, and ΔH in the thermodynamic properties. ΣDP 12–24 was significantly positively correlated with HPV, CPV, CSV, PeT, and PaT in the RVA spectrum characteristic values and T_O_, T_P_, T_C_, and ΔH in the thermodynamic properties. ΣDP 25–36 was significantly negatively correlated with PaT in the RVA spectrum characteristic values and T_O_, T_P_, and T_C_ in the thermodynamic properties. No correlation was present between ΣDP ≥ 37 and the physicochemical and thermodynamic properties of each starch sample. The average DP was only significantly positively correlated with PeT and PaT in the RVA spectrum characteristic values, indicating that the lesser the short-chain distribution of glutinous rice amylopectin (DP 6–11), the more the medium-long-chain distribution (DP 12–24), and the lesser the long-chain distribution (DP 25–36), the higher the RVA spectrum characteristic values, this is consistent with Zhou’s study [20]. The higher the average DP of the glutinous rice, the higher the pasting time and PaT of starch. No significant correlation was present between the chain-length distribution of glutinous rice amylopectin and the relative crystallinity of starch, indicating that the complexity of the amylopectin structure did not affect the crystallinity of the glutinous rice starch granules.

## 4. Discussion

### 4.1. Amylopectin Chain Length and Its Distribution in the Glutinous Rice Varieties

Amylopectin is the main component of the rice endosperm. With the development of science and technology, the methods for amylopectin structure determination are also gradually developing. Presently, these methods can be divided into two categories: electrophoresis and chromatography, with each category consisting of various submethods and having its own advantages and disadvantages. The methods can be divided into FACE, enzymatic method, gel chromatography [13], and spectrophotometry [21]. Umemoto et al. [22,23] believed that the distribution of amylopectin short-length chains with DP ≤ 11 and the medium-length chains with DP 12–24 in different rice varieties was relatively different, whereas the number of long-length chains with DP ≥ 25 was the same. Nakamura et al. [24] used the FACE method based on capillary electrophoresis to determine the amylopectin structures of 129 different rice varieties. The amylopectin structures were divided into long-chain type (L-type) and short-chain type (S-type), and the amylopectin chain ratio (ACR) of ΣDP ≤ 10/ΣDP ≤ 24 of the L-type amylopectin was less than 0.20, whereas that of ΣDP ≤ 10/ΣDP ≤ 24 of the S-type amylopectin was greater than 0.24.

He et al. [9] used the FACE method based on a DNA sequencer to determine the amylopectin structure of 50 different indica and japonica rice varieties. Based on the actual differences in the chain length and chain-length distribution in different types of rice varieties, the ACR of ΣDP ≤ 11/ΣDP ≤ 24 was used as the classification basis. All amylopectin varieties were divided into two types: type I and type II. The ACR of type I amylopectin was less than 0.22, corresponding to the L-type amylopectin, and that of type II amylopectin was greater than 0.26, corresponding to the S-type amylopectin. In this study, HAPED-PAD [5], using a Thermo ICS5000+ ion chromatography system equipped with pulsed amperometric detection, was performed to determine amylopectin chain length distribution in the 29 glutinous rice varieties. The HAPED-PAD method is simpler and more reproducible than other methods, such as the enzymatic method, spectrophotometry, and FACE, as it can accurately determine the distribution of amylopectin chain lengths with different DP and can more effectively analyze the correlation between amylopectin chain length distribution and the physicochemical properties of starch. The results showed that except for Heixiangnuo and Zaonuo116, whose ACR were 0.231 and 0.236, respectively, the ACR of other glutinous rice varieties were greater than 0.26 and ranged from 0.301 to 0.324, which was consistent with the results of He et al. [9], thus verifying the accuracy of the HAPED-PAD method.

### 4.2. Correlation between the Amylopectin Structure and Physicochemical Properties of Starch

GT refers to the temperature at which a large number of water-absorbing starch granules undergo irreversible expansion, birefringence, and crystallinity disappearance after being heated in a suspension aqueous solution [25]. Vandeputte et al. [2] reported that ΣDP 6–9 was negatively correlated with GT, whereas ΣDP 12–22 was positively correlated with GT. Qi et al. [26] analyzed the starch chain length distribution in six glutinous rice varieties and found that the higher the ratio of ΣDP 13–24, the higher the GT. Satoh et al. [27] reported a decrease in starch extra-long and long chains in the *ae* rice mutant and a significant increase in the short chains, resulting in a significant decrease in T_O_ of starch. He et al. [9] found that the short chains with DP 6–11 and medium-length chains with DP 13–24 were significantly negatively and significantly positively correlated with pasting temperature (PT), respectively. The long branch chains with DP 28–34 were significantly negatively correlated with GT, and the extra-long branch chains with DP 39–49 were significantly positively correlated with it in all of the varieties. In this study, T_O_, T_P_, and T_C_ of the test varieties were significantly negatively correlated with ΣDP 6–11 and ΣDP 25–36 and significantly positively correlated with ΣDP 12–24. These results were consistent with those of previously reported studies, indicating that the amylopectin structure plays similar and important role in the GT of glutinous rice and non-glutinous rice. The relative number of short chains with DP 6–11 and long chains with DP 25–36 should be increased, whereas the relative number of medium-length chains with DP 12–24 should be reduced to improve the rice amylopectin structure for reducing GT.

The RVA spectrum characteristic values are closely related to the eating quality of rice. The rice varieties with lower HPV, CPV, SBV, CSV, PeT, and PaT and a higher BDV are considered to have better grain quality, softer texture, better viscosity, and cold rice texture [9]. By performing gel chromatography, Cai et al. [7] found that FrIII in the amylopectin structure was significantly positively correlated with PKV and BDV in the RVA spectrum, whereas FrI and FrII were negatively correlated with it. Han et al. [8] used the gel chromatography and determined that FrI was significantly negatively correlated with BDV, whereas FrIII was significantly positively correlated with it. By performing chromatography, Jin et al. [28] found that the amylopectin FrI + FrII content was significantly positively correlated with HPV, CPV, SBV, and CSV and was significantly negatively correlated with BDV. He et al. [9] suggested that the ACR of short chains with DP 6–11 and medium-length chains with DP 13–24 had no significant correlation with the RVA spectrum characteristic values in the low-AC rice varieties, whereas in the high-AC rice varieties, SBV was positively correlated with ΣDP 6–11 and negatively correlated with ΣDP 13–24. ΣDP 28–34 and ΣDP 39–49 also had no significant correlation with the RVA spectrum characteristic values in the general rice varieties. Among the 29 glutinous rice varieties used in this study, the distribution of short chains with DP 6–11 was significantly negatively correlated with PKV, HPV, CPV, CSV, PeT, and PaT. The distribution of medium-long chains with DP 12–24 was significantly positively correlated with HPV, CPV, CSV, PeT, and PaT. The long chains with DP 25–36 were significantly negatively correlated with PaT. DP was significantly positively correlated with PeT and PaT, which was inconsistent with the results of previous studies using non-glutinous rice varieties. The reason behind this is glutinous rice is considerably different from non-glutinous rice, as it almost does not contain amylose. The difference in the amylopectin structure leads to differences in the physicochemical properties of starch among different varieties. Therefore, in glutinous rice varieties, the distribution and proportion of amylopectin short and medium-long chains are important factors in determining their eating quality.

## 5. Conclusions

In this study, the amylopectin chain ratio ΣDP ≤ 11/ΣDP ≤ 24 of 29 glutinous rice varieties was greater than 0.26, signifying that these varieties contained type II amylopectin. The results of the correlation analysis showed that ΣDP 6–11 was significantly negatively correlated with T_O_, T_P_, and T_C_ among the thermodynamic properties, ΣDP 12–24 was significantly positively correlated with To, Tp, and Tc, ΣDP 25–36 was significantly negatively correlated with To, Tp, and Tc. ΣDP 6–11 was significantly negatively correlated with HPV, CPV, CSV, PeT, and PaT among the RVA profile characteristics, ΣDP 12–24 was significantly positively correlated with HPV, CPV, CSV, PeT, and PaT. Therefore, we concluded that the amylopectin structure had a greater effect on the T_O_, T_P_, T_C_, ΔH and peak viscosity, HPV, CPV, CSV, PeT, and PaT. The glutinous rice varieties with a higher distribution of short chains and a lower distribution of medium and long chains in the amylopectin structure resulted in lower GT and RVA spectrum characteristic values. These new findings will significantly assist in revealing the genetic basis of the formation of amylopectin structure in glutinous rice and provide a theoretical basis for the breeding and improvement of glutinous rice starch quality.

## Figures and Tables

**Table 1 foods-12-00460-t001:** Amylose content determination results of the 29 glutinous rice varieties.

Number	Variety	AC (%)	PCR Product Size	Number	Variety	AC (%)	PCR Product Size	Number.	Variety	AC (%)	PCR Product Size
1	Heinuomi	1.76	492 bp	11	Yunhenuo	2.91	492 bp	21	Bancangxiangnuo	2.43	492 bp
2	Wangdingnuo	1.94	492 bp	12	Chuangengnuo	2.77	492 bp	22	Lirenzi	2.72	492 bp
3	Hengniannuo	2.21	492 bp	13	Anyinuo	0.46	492 bp	23	Zhongkeheinuo1	2.31	492 bp
4	Xiangyanuo	2.33	492 bp	14	Heixiangnuo	2.32	492 bp	24	Shinuo	1.92	492 bp
5	Hongkenuo	1.30	492 bp	15	Guazixiangdao	2.34	492 bp	25	Yangkenuo	2.27	492 bp
6	Baishanuo	2.37	492 bp	16	Yuyangnuo	2.87	492 bp	26	3010-1	2.17	492 bp
7	Guangxiannuo	2.22	492 bp	17	Zaonuo116	2.23	492 bp	27	Zibaoxiangnuo	1.45	492 bp
8	Liuyangnuo	2.43	492 bp	18	Hongheinuo	2.12	492 bp	28	Hongrangheinuo	0.89	492 bp
9	Wannuo53	0.94	492 bp	19	Maobinuo	2.53	492 bp	29	Fuwannuo18	0.48	492 bp
10	Zaoxiannuo	2.52	492 bp	20	Xixiangnuo	2.47	492 bp				

**Table 2 foods-12-00460-t002:** Distribution of amylopectin chain length in the 29 glutinous rice varieties.

Number	Name	Type	DPn	Amylopectin Chain Length Distribution/%			
				DP ≤ 11	DP 12–24	DP 25–36	DP ≥ 37
1	Heinuomi	*indica*	19.488	24.222	54.075	11.389	10.315
2	Wangdingnuo	*indica*	19.921	23.633	53.206	12.076	11.086
3	Hengniannuo	*indica*	19.179	24.949	54.506	11.249	9.596
4	Xiangyanuo	*indica*	19.695	24.693	52.609	12.066	10.686
5	Hongkenuo	*japonica*	19.417	24.028	54.427	11.488	10.057
6	Baishanuo	*japonica*	19.588	23.793	54.150	11.630	10.426
7	Guangxiannuo	*japonica*	19.591	23.892	54.092	11.557	10.459
8	Liuyangnuo	*indica*	19.145	25.255	53.920	11.310	9.514
9	Wannuo53	*indica*	19.604	23.778	54.191	11.494	10.537
10	Zaoxiannuo	*japonica*	19.163	25.007	54.263	11.081	9.649
11	Yunhenuo	*indica*	19.297	24.515	54.275	11.400	9.809
12	Chuangengnuo	*japonica*	19.318	24.054	54.617	11.570	9.759
13	Anyinuo	*indica*	19.273	24.365	54.436	11.544	9.655
14	Heixiangnuo	*japonica*	20.032	18.078	60.171	11.087	10.663
15	Guazixiangdao	*japonica*	19.011	25.332	54.380	10.861	9.427
16	Yuyangnuo	*indica*	19.538	23.519	54.598	11.706	10.177
17	Zaonuo116	*japonica*	19.663	18.695	60.592	10.982	9.732
18	Hongheinuo	*indica*	20.020	23.408	53.118	12.231	11.242
19	Maobinuo	*japonica*	19.456	23.742	54.781	11.228	10.249
20	Xixiangnuo	*indica*	19.691	23.379	54.365	11.690	10.566
21	Bancangxiangnuo	*indica*	19.109	24.787	54.582	11.138	9.494
22	Lirenzi	*indica*	20.033	23.466	52.983	12.251	11.300
23	Zhongkeheinuo1	*indica*	19.392	25.214	52.708	11.934	10.144
24	Shinuo	*indica*	19.120	24.887	54.332	11.390	9.391
25	Yangkenuo	*japonica*	19.547	23.761	54.435	11.305	10.500
26	3010-1	*indica*	19.343	24.399	54.203	11.508	9.890
27	Zibaoxiangnuo	*japonica*	19.900	23.229	53.771	11.847	11.153
28	Hongrangheinuo	*indica*	19.295	25.000	53.736	11.269	9.995
29	Fuwannuo18	*japonica*	19.209	25.145	53.597	11.393	9.864

**Table 3 foods-12-00460-t003:** The Rapid Visco Analyzer spectrum values of the 29 glutinous rice varieties.

Number	PKV/cP	HPV/cP	BDV/cP	CPV/cP	SBV/cP	CSV/cP	PeT/min	PaT/°C
1	3076	1603	1473	1874	−1202	271	3.87	73.98
2	2239	1226	1014	1455	−784	229	3.84	72.37
3	2482	1424	1058	1683	−799	259	3.84	71.83
4	2472	1372	1100	1631	−841	259	3.82	72.07
5	2910	1728	1182	2097	−813	369	4.11	73.47
6	1777	1118	660	1368	−409	250	4.09	75.02
7	2938	1512	1425	1850	−1088	338	4.00	72.33
8	2940	1535	1406	1873	−1067	339	3.76	71.83
9	2815	1662	1153	2099	−716	437	4.02	72.57
10	2805	1554	1251	1864	−941	309	3.96	72.95
11	1831	1179	652	1467	−364	288	3.93	73.15
12	2068	1283	786	1532	−536	250	4.03	74.33
13	2268	1343	925	1600	−668	257	4.04	73.92
14	3425	2111	1314	2563	−862	452	4.78	83.00
15	2029	1062	967	1292	−737	230	3.76	73.18
16	2701	1674	1027	2047	−654	373	4.13	73.97
17	3095	2052	1043	2536	−559	484	4.76	81.93
18	2825	1498	1328	1832	−994	334	3.89	72.28
19	2260	1473	787	1738	−522	265	4.09	75.32
20	2826	1532	1294	1848	−978	316	4.20	72.83
21	2989	1594	1395	1910	−1080	315	3.87	72.38
22	3087	1649	1437	1978	−1109	328	3.91	72.88
23	2930	1669	1261	2048	−882	379	4.07	72.62
24	1523	882	641	1085	−438	203	3.76	70.95
25	2116	1205	911	1442	−675	236	3.93	74.22
26	2340	1347	993	1593	−747	246	4.07	73.13
27	2225	1259	966	1494	−731	235	3.89	73.42
28	3107	1578	1529	1884	−1223	306	3.87	73.42
29	1877	1220	658	1470	−408	250	4.04	73.15

**Table 4 foods-12-00460-t004:** The thermodynamic characteristic values of the 29 glutinous rice varieties.

Number	To/°C	Tp/°C	Tc/°C	ΔH/J·g^−1^	RC/%
1	64.62	71.76	80.25	10.44	38.08
2	63.07	71.00	79.49	7.70	35.42
3	62.78	70.91	79.51	7.58	35.24
4	63.39	71.04	79.68	7.69	35.76
5	63.35	72.51	82.77	9.73	33.83
6	65.18	71.56	80.51	9.95	31.75
7	63.86	71.50	80.50	8.87	36.63
8	63.21	70.34	77.96	10.90	36.63
9	63.69	70.72	80.55	7.89	34.64
10	65.04	73.56	88.18	9.20	38.85
11	63.59	71.28	81.86	8.64	37.12
12	64.56	71.69	80.47	9.95	37.25
13	64.63	72.05	80.78	9.67	36.65
14	75.37	79.97	87.11	12.10	34.63
15	65.09	70.25	77.92	8.50	36.02
16	64.32	71.17	79.66	10.17	33.84
17	76.91	80.66	87.02	11.89	36.09
18	63.71	70.33	79.27	8.28	32.48
19	66.53	72.98	80.38	9.10	34.59
20	64.74	71.86	79.45	6.33	33.00
21	64.64	71.08	79.70	7.49	35.86
22	62.84	70.70	79.54	11.30	40.28
23	63.58	70.95	80.16	10.21	36.15
24	62.54	70.50	79.86	6.86	33.87
25	64.68	72.92	82.35	11.83	38.71
26	64.32	71.64	80.59	8.10	36.05
27	64.59	71.23	79.51	7.63	38.09
28	64.66	71.89	80.03	11.30	40.47
29	63.70	70.68	78.85	10.29	35.71

**Table 5 foods-12-00460-t005:** Correlation analysis of the physicochemical properties of starch.

Parameters	PKV	HPV	CPV	BDV	SBV	CSV	PeT	PaT	RC
To	0.363	0.609 **	0.614 **	0.048	0.117	0.584 **	0.882 **	0.965 **	−0.066
Tp	0.394 *	0.643 **	0.642 **	0.070	0.093	0.585 **	0.891 **	0.944 **	0.003
Tc	0.318	0.523 **	0.527 **	0.054	0.086	0.499 **	0.684 **	0.689 **	0.117
ΔH	0.353	0.464 *	0.459 *	0.176	−0.077	0.403 *	0.446 *	0.591 **	0.394 *

Note: * indicates a significant correlation at the 0.05 level, and ** indicates a very significant correlation at the 0.01 level.

**Table 6 foods-12-00460-t006:** Correlation analysis between the amylopectin structure and physicochemical properties of starch.

	Parameters	DPn	Amylopectin Chain Length Distribution (%)			
			DP ≤ 11	DP 12–24	DP 25–36	DP ≥ 37
RVA spectrum characteristic values	PKV	0.333	−0.399 *	0.305	−0.003	0.224
	HPV	0.363	−0.632 **	0.568 **	−0.108	0.161
	CPV	0.360	−0.638 **	0.577 **	−0.110	0.151
	BDV	0.240	−0.088	−0.018	0.103	0.246
	SBV	−0.172	−0.084	0.188	−0.145	−0.243
	CSV	0.319	−0.623 **	0.575 **	−0.110	0.102
	PeT	0.369 *	−0.883 **	0.864 **	−0.251	0.068
	PaT	0.347 *	−0.908 **	0.919 **	−0.386 *	0.050
thermodynamic properties	To	0.282	−0.898 **	0.951 **	−0.454 *	−0.030
	Tp	0.287	−0.886 **	0.935 **	−0.438 *	−0.023
	Tc	0.142	−0.629 **	0.705 **	−0.407 *	−0.086
	ΔH	0.118	−0.401 *	0.419 *	−0.217	−0.004
	RC	−0.115	0.169	−0.128	−0.104	−0.036

Note: * indicates a significant correlation at the 0.05 level, and ** indicates a very significant correlation at the 0.01 level.

## Data Availability

The data that support the findings of this study are available from the corresponding author upon reasonable request.

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
