# Peer review of "Research on the Relationship between the Amylopectin Structure and the Physicochemical Properties of Starch Extracted from Glutinous Rice"

_foods, 2023, doi:10.3390/foods12030460_

Round 1

Reviewer 1 Report

Comments(No.2129313-Foods)

This manuscript titled “Research on the relationship between the amylopectin structure and the physicochemical properties of starch extracted from glutinous rice”determined the correlation between amylopectin structure and the physicochemical properties of starch extracted from 29 glutinous rice varieties. This work is interesting for practice, butsome points have to addressed for authors’ consideration. Thus, authors should make a major modification to their manuscript before resubmitting to this journal.

Specific points are issued as follows.

1/Ln 78, do the samples need to be stored for three months?

2/Ln 84, the rice flour was dried firstly and placed at room temperature for 24 h to maintain its water content. This is confusing, and the water content of all samples are same?

3/Ln 102, all methods and determinations should be presented simply for reading clearly.

4/Ln 103, “Chinese national standard GBT24852-2010” is missing in the section reference.

5/Ln 128, how many repetitions were done in this work, and all data should be supplemented with significant difference.

6/Ln 137-138, references are necessary. The data in table 1 is not enough to confirm this result. Although the AC is low, the correlation should be explored further.

7/Ln 144-147, references are necessary for supporting this expression. Additionally, where are the results of Wx genotyping in table 1? Otherwise, what is the meaning for the section 1.2. Wx genotyping?

8/Ln 216-219, references are necessary for supporting this expression.

9/Tables, all date is presented as tables not figures, thus it is different to find the results. The table 2 and 3 should be classified as several groups to present the findings.

10/Ln 240-245, it should not be here. Additionally, there is no data to compare the accuracy of different methods. Thus, the part 3.1 is no meaningful. 

11/Ln 263-267, authors suggest that HAPED-PAD method is simple and good tool, but there are samples obtained randomizedly to confirm the accuracy of this method?

12/ The section conclusion should be supplemented.

13/For the part References, the references should be refreshed, and the journal name should be unified for their format.

Reviewer 2 Report

This paper is focused on the research on the relationship between the amylopectin structure and the physicochemical properties of starch extracted from glutinous rice.

Although the manuscript is well presented, I suggested to the authors to introduce some minor changes to improve the paper. The authors need to provide or add a wider description of involving glutinous rice as a material for determining the structure and physicochemical properties of amylopectin which can be viewed from several aspects such as economic, practicality, and several other important aspects.

The authors needs to confirm and clarify about replication for this research. If this research has been replicated, the author needs to provide or add information about the number of replications that have been carried out.

Where is ‘Conclusions’ chapter? It should be supplemented.

General information concerning Literature Citation should be as recommended in Instructions for Authors (https://www.mdpi.com/journal/foods/instructions); Literature citation additionally requires improvement and correction;

Reviewer 3 Report

Good work very informative.

1. For the benefit of the reader, the authors need to define glutinous rice and how different from common rice.  Is it the type of rice containing low or NO amylose? If so, please clarify

2. In the abstract, the appreciations need to be defined clearly.

3. Line 84 why the drying process, what was the moisture content before and after drying.  This can be significant for the milling process because it may damage the starch if it was too low.

4. Method 1.3 and 1.4, the authors need to add a brief description of the method

5. Method 1.5, HPLC conditions, pH, moving phase …….

6. Method 1.7.2, DSC conditions such as sample weight, moisture content, and equilibration time

7. Statistics, please state the experimental design used.

8. Line 144 and 159, these statements should be in the discussion section.

9.  Line 159-162, please present a reference.

10. Lines 173-178 need a reference

11. Line 182 -184, please add a different justification for the variation, because this justification is supported by the data presented here.

Round 2

Reviewer 1 Report

No more comments.

Author Response

Dear Editor Ms. Alexandra-Madalina Mateescu and Reviewer,

Thanks very much for taking your time to review this manuscript. I really appreciate all your comments and suggestions! Thanks again!